# Syn-QL: Prefernce Optimization with Synthetic Data for Text-to-SQL

**Ruilin Hu**
2024210886
Tsinghua University
hrl24@mails.tsinghua.edu.cn

**Lu Fan**
2024210883
Tsinghua University
fanl24@mails.tsinghua.edu.cn

**Yizhe Chen**
2023311175
Tsinghua University
chenyizh23@mails.tsinghua.edu.cn

## Abstract

This paper addresses the challenge of improving the performance of open-source Large Language Models (LLMs) in Text-to-SQL tasks, where a natural language query is converted into an SQL statement for database interaction. Despite their accessibility and cost-efficiency, open-source LLMs lag behind closed-source models in accuracy. To bridge this gap, we introduce Syn-QL, a framework leveraging synthetic data generation and self-training techniques to fine-tune models iteratively. Our method utilizes a dual-model approach, pairing a SQL Writer and SQL Verifier to enhance the quality of SQL outputs through repeated refinement. Experimental results demonstrate notable performance improvements on established benchmarks, including Spider and BIRD, underscoring Syn-QL's potential to make open-source LLMs more competitive in Text-to-SQL tasks.

## 1 Intro

### 1.1 Problem Definition

The Text-to-SQL task aims at converting a user's natural language(NL) question into a valid SQL query that can be executed in a given database. Text-to-SQL allows users without SQL proficiency to interact with databases, perform data exploration, make data-driven decisions, and build the initial version of more complex SQL queries[10, 5]. Located at the intersection of data management [1, 2] and natural language processing (NLP) [11, 13], Text-to-SQL has received considerable attention.

**Text-to-SQL Task.** Assume $Q$ is a natural language question and $D = \{S, V\}$ is a relational database where $S$ is the database schema and $V$ is the collection of all the values in the database. Given $D$, the task of Text-to-SQL is to generate an SQL query $S$ to answer the user's natural language question $Q$. The generated SQL query $S$ is considered correct iff $S$ can be successfully executed in the database $D$ and retrieve the correct content.

**LLM for Text-to-SQL.** Given the impressive capabilities of Large Language Models(LLMs) in language understanding and code generation, numerous studies have endeavored to apply LLMs to the Text-to-SQL task [7, 3, 9]. Although high accuracy was obtained, these prior methods depend heavily on the capability of powerful close-source LLMs, which have unignorable impacts on practical applications. For example, many methods based on few-shots prompting require inputting a large

37th Conference on Neural Information Processing Systems (NeurIPS 2023).

number of tokens into GPT-4, which can incur significant economic costs. Additionally, it is often unacceptable for companies to send proprietary data to LLM service providers.

## 1.2 Motivation

**Open-Source LLMs for Text-to-SQL.**  Open-source large language models for the Text-to-SQL task offer advantages such as low latency and reduced costs. However, they exhibit a significant performance gap compared to their closed-source counterparts. In particular, the popular coding LLM StarCoder-15B still demonstrates a 30% lower execution accuracy on the challenging BIRD benchmark [5].

**Training Open-Source Text-to-SQL LLMs.**  Given the substantial disparity between open-source and closed-source models in Text-to-SQL, numerous aspects remain unexplored. We focus on enhancing the capability of open-source base models with fine-tuning methods.

**Challenges.**  Text-to-SQL can be considered a special form of code generation, with additional contextual information including the database schema and potentially external knowledge. But compared to plain code generation, the Text-to-SQL task requires models to understand probably unseen and complex database schemas and possess strong reasoning capability to generate an matching SQL query. Beyond its inherent challenges, improving open-source LLMs through supervised fine-tuning encounters several obstacles.

**Challenge 1: High difficulty in reasoning.**  SQL syntax is intricate. Even experienced data scientists often need to engage in trial-and-error processes to formulate the correct SQL queries. Consequently, the Text-to-SQL task necessitates models with strong reasoning capabilities, challenging models to interpret natural language queries, understand the underlying database schema, and generate syntactically correct and semantically accurate SQL queries that correspond to the user's intent. While Chain-of-Thought (CoT) [12] is efficient in enhancing the reasoning capabilities of LLMs, it faces limitations in the context of SQL query generation. SQL queries exhibit strong interdependencies between their components, where the formulation of earlier parts often requires consideration of subsequent elements. This characteristic challenges the linear, front-to-back output approach typically associated with CoT methods.

**Challenge 2: Limited Training Data.**  Text-to-SQL faces another issue of scarce high-quality training data. To empower models with cross-domain generalization capabilities, a diverse array of database schemas is essential. Concurrently, to enable models to generate complex queries, a wide variety of natural language questions and their corresponding SQL answers is required. However, in reality there is a significant scarcity of diverse and semantically aligned (Schema, NL Question, SQL) pairs. The acquisition of high-quality training data relies on manual expert annotation, which is time-consuming and costly.

## 2  Method

We aim to investigate methods for enhancing the Text-to-SQL capabilities of open-source Large Language Models (LLMs) through the utilization of synthetic data generation and self-training techniques.

Specifically, our approach leverages the concept of self-training, where the model's own outputs are utilized iteratively as training data for fine-tuning.Following the initial fine-tuning phase, we continue to input synthetically generated questions into the model. Subsequently, we extract high-quality question-answer pairs from these outputs to serve as additional data for further fine-tuning iterations. This process aims to create a positive feedback loop, potentially enhancing the model's performance with each cycle.

To address the challenge of noisy outputs resulting from synthetic input questions, we have designed and implemented a verification mechanism using a separate LLM, which we term the SQL Verifier. This additional model serves as a semantic validator, tasked with assessing the correspondence between NL queries and their associated SQL outputs.While our research indicates that the Chain-of-Thought approach proves challenging to implement directly in the Text-to-SQL generation process,

we have discovered its potential for enhancing the verification phase. Building upon this insight, we propose a novel method termed Chain-of-Clause (CoC) to augment our verification LLM's capabilities.

We propose an iterative optimization framework leveraging two specialized Large Language Models (LLMs) to enhance the quality and reliability of Text-to-SQL translations. This dual-model approach consists of a SQL Writer and a SQL Verifier, working in tandem to refine the model's performance through successive iterations.

## 2.1 Verifier Training

---

**Algorithm 1** Generating SQL Verifier Model

---

**Input:** pre-trained LLM $M_{pretrain}$, training dataset $D$, annotator LLM $M_{ant}$
**Output:** fine-tuned verifier model $M_{verify}$
1: Split $D \Rightarrow D_{sft}, D_{sample}$          ▷ *Split D into sft set and sample set*
2: Fine-tune $M_{pretrain}$ with $D_{sft} \Rightarrow M_{sft}$
3: Execute $D_{sample}$ with $M_{sft} \Rightarrow \{(q_i, g\_sql_i, p\_sql_i)\}_{i=1}^{L}$      ▷ *Sample multiple sql queries*
4: Judge correctness with $\{gold\_sql_i\}_{i=1}^{L} \Rightarrow \{(q_i, p\_sql_i, c_i)\}_{i=1}^{L}$
5: Annotate Chain-of-Clauses rationales using $M_{ant} \Rightarrow \{(q_i, p\_sql_i, c_i, coc_i)\}_{i=1}^{L}$
6: Filter data if CoC matches correctness $\Rightarrow V = \{(q_i, p\_sql_i, coc_i)\}_{i=1}^{K}$
7: Fine-tune $M_{pretrain}$ with $V \Rightarrow M_{verify}$
8: **return** SQL verifier model $M_{verify}$

---

# 3 Experiment

## 3.1 Evaluation Benchmarks

We evaluate the effectiveness of our method with recognized Text-to-SQL benchmarks across multiple datasets.

**General Benchmark**    Spider[14] is a widely-recognized Text-to-SQL benchmark. Spider contains 7000 human-annotated Text-to-SQL pairs in its training set and 1034 pairs in the validation set, across 200 different databases and 138 domains.

**Challenging Benchmark**    BIRD[5] is a challenging benchmark of large real-world databases. BIRD contains 95 databases across 37 fields and 9428 high-quality Text-to-SQL pairs. BIRD features massive and dirty database contents and requires Text-to-SQL systems to reason on external expert knowledge to generate SQL queries.

## 3.2 Evaluation Metrics

We report two common evaluation metrics: Exact Match Accuracy (EM) and Execution Accuracy (EX). EM requires every subcomponent of the predicted SQL query to match the reference SQL query in the dataset. EX determines equivalence between a predicted SQL query and a reference SQL query if they produce identical results across various database instances. EX is considered a more accurate measurement of Text-to-SQL methods since multiple correct SQL queries can differ in output style.

## 3.3 Evaluation Results

This is a placeholder, see Table1.

# 4 Related Work

**LLM based Text-to-SQL**    Recently, many studies focus on utilizing LLMs to solve Text-to-SQL tasks by developing novel pipelines and exploring innovative prompting techniques. For instance,

| Model / Method | | Spider | | BIRD | |
| --- | --- | --- | --- | --- | --- |
| **Name** | **# Calls** | **Dev** | **Test** | **Dev** | **Test** |
| **Prompting Methods w/ Closed-Source LLMs** | | | | | |
| GPT-4 | 1 | 72.9 | - | 49.2 | 54.9 |
| DIN-SQL + GPT-4 | 4 | 82.8 | 85.3 | 50.7 | 55.9 |
| DAIL-SQL + GPT-4 | 4 | 83.5 | 86.2 | 54.8 | 57.4 |
| TA-SQL + GPT-4 | 3 | 85.0 | - | 56.2 | 59.1 |
| PTD-SQL + GPT-4 | 3 | 85.7 | - | 57.0 | - |
| **Fine-tuning Open-Source LLMs** | | | | | |
| SENSE-7B | 1 | 83.2 | 83.5 | 51.8 | 59.3 |
| SENSE-13B | 1 | 84.1 | 86.6 | 55.5 | 63.4 |
| SFT CodeS-7B | 1 | 85.5 | - | 55.8 | 60.3 |
| SFT CodeS-15B | 1 | 85.4 | - | 57.2 | 59.2 |
| Qwen 2.5-Coder-7B | 1 | 82.0 | - | 51.1 | - |
| **Multi-Stage NL2SQL w/ Open-Source LLMs** | | | | | |
| Ours-7B | 2 | **87.6** | – | **63.0** | – |

Table 1: Performance Comparison of Different Models

ACT-SQL[15] generates Chain-of-thought prompts automatically , DIN-SQL[7] decomposes complex problems into LLM-solvable easy problems, DAIL-SQL[3] samples semantically similar examples as prompts to improve Text-to-SQL performance, PTD-SQL[6] partitions examples in different banks and selects shots as examples at run-time. Li et al.[4] conducted sufficient experiments about the pros and cons of each popular method and used genetic algorithms to propose SuperSQL, which chooses the best option at different Text-to-SQL stages. Apart from mainly employing proprietary LLMs to solve Text-to-SQL tasks, ZeroNL2SQL[2] combines small pretrained models as SQL skeleton parsers and utilizes LLMs to obtain the complete SQL, and DTS-SQL[8] fine-tunes two open-source LLMs separately to prune the database schemas and generate SQL queries.

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
