# OpenReview forum: "Syn-QL: Prefernce Optimization with Synthetic Data for Text-to-SQL"
_tsinghua.edu.cn/THU/2024/Fall/AML — THU 2024 Fall AML Submission_

### Official Review · ~Guilherme_Félix_Diogo1 · 2024-11-06
**Clear explanation of the problem but the methods are a little incomplete**

**Rating:** 8
**Confidence:** 4

**Review:**

The proposal suggests a feasible method for improving the performance of Text-to-SQL systems for open-source LLMs via synthetic data generation and dual model architecture. The project is relevant in making clear the performance difference between open-source models closed-source ones, regarding especially complex SQL generation tasks. The idea of looping back and forth between an SQL Writer and an SQL Verifier is innovative. The only negative aspect is that the methodology section of the proposal is too general and does not provide enough details to understand how the Chain-of-Clause method is fitted into the existing CoT method. Providing some more details would enhance understanding.

---

### Official Review · ~Chua_Shei_Pern1 · 2024-11-06
**Innovative proposed method**

**Rating:** 9
**Confidence:** 4

**Review:**

The proposal introduces Syn-QL which is a framework that leverages synthetic data generation and iterative self-training to improve Text-to-SQL performance for open-source language models. While the dual-model approach is innovative, adding more details on the SQL Verifier's validation mechanisms could strengthen the proposal's comprehensiveness and implementation clarity.

---

### Official Review · ~Ziyi_Liu9 · 2024-11-07
**Well Done**

**Rating:** 9
**Confidence:** 3

**Review:**

There is clear logic in the motivation, problem definition, challenges, and corresponding solutions. The approach is also innovative in leveraging an SQL Verifier to ensure the correctness of the generated output. The proposal would benefit from a more detailed description of the CoC method.

---

### Official Review · ~Tim_Bakkenes1 · 2024-11-09
**Great proposal but fails to meet page limit**

**Rating:** 7
**Confidence:** 4

**Review:**

This is a great proposal, although it fails to meet the requirements.

The problem definition is very clear and the prospect of a reliable natural language interface for interacting with SQL databases is very impactful. The proposal also includes a formal definition which makes it easier for the reader to understand. The motivation for your selected research area is also well written.

The method is innovative and includes both a SQL Writer and a SQL Verifier that will verify the output of the SQL writer. The algorithm for the writer is also well documented and you clearly state how your model will be evaluated.

There is a lot of good in your proposal. The main issue is that it is four pages long, excluding references. That is twice the allowed amount of two pages. The contents of your proposal has to be condensed down to fit the page limit.

---

### Official Review · ~Zihan_Yan2 · 2024-11-10
**A good proposal, but it exceeds the page limit**

**Rating:** 7
**Confidence:** 4

**Review:**

The overall structure of this proposal is clear, with a well-defined problem statement and a detailed technical approach, even including preliminary experimental results. However, the proposal does not meet the format requirements, as it spans four pages.

---

### Official Review · ~Rosalie_Butte1 · 2024-11-10

**Rating:** 8
**Confidence:** 4

**Review:**

The paper proposes a method to improve Text-to-SQL through synthetic data generation and self-training.

It clearly outlines the background and challenges. It also shows a well-structured and innovative approach of the implementation of a dual-model. However, a more detailed explanation of the Chain-of-Clause method could further improve the overall understanding. It also exceeds the page limit.

---

### Official Review · ~Kehan_Zheng1 · 2024-11-11
**Good Proposal**

**Rating:** 8
**Confidence:** 4

**Review:**

This proposal defines the problem with clarity and precision, emphasizing the need and current challenges in enhancing Text-to-SQL performance for open-source LLMs. The proposed framework is well-structured, presenting a comprehensive method with promising experimental results, demonstrating its potential effectiveness.

However, there are a few concerns. Firstly, the proposal far exceeds the 2-page limit. Additionally, as a proposal, the inclusion of a fully developed method may be premature, while outlining preliminary concepts and future work directions might be more appropriate.

---

### Official Review · ~Tianxing_Yang1 · 2024-11-11
**Review of Syn-QL Proposal for Enhancing LLM in Text-to-SQL Tasks**

**Rating:** 7
**Confidence:** 3

**Review:**

This proposal introduces Syn-QL, a method that leverages data synthesis and validation to iteratively fine-tune large language models (LLMs), with the goal of improving performance in text-to-SQL tasks.

Strengths:

- The use of LLMs for text-to-SQL tasks is a practical and relevant area of research with significant potential applications.
- The proposal outlines a clear methodology and provides preliminary experimental results to support its feasibility.

Areas for Improvement:

- The length of the proposal exceeds the page limit (2 pages).

---

### Official Review · ~Shuangyue_Geng1 · 2024-11-11
**Great proposal, but could be clearer and more concise**

**Rating:** 8
**Confidence:** 4

**Review:**

The paper presents a well-structured and innovative approach to improving the performance of open-source Large Language Models (LLMs) in Text-to-SQL tasks. The logical flow of the content is clear, making it easy to follow the authors' motivation, methodology, and results. However, there are some areas for improvement. The paper lacks detailed explanations of the symbols used in the algorithms and tables, making parts of the methodology challenging to understand. Additionally, the paper exceeds the two-page limit.

---

### Official Review · ~Isak_Tønnesen1 · 2024-11-12
**Review: Prefernce Optimization with Synthetic Data for Text-to-SQL**

**Rating:** 8
**Confidence:** 4

**Review:**

This proposal presents Syn-QL, an innovative framework for improving open-source LLMs' performance on Text-to-SQL tasks. The approach is well-structured, combining synthetic data generation with a novel dual-model architecture (SQL Writer and SQL Verifier) for iterative refinement. The proposal has a clear problem formulation, comprehensive understanding of current limitations in Text-to-SQL systems, and technically sound methodology. The preliminary results on Spider and BIRD benchmarks are particularly impressive, showing substantial improvements over existing methods. While the proposal exceeds the page limit and could better detail the Chain-of-Clause method, its strong technical foundation and potential impact on making open-source LLMs more competitive in this space make it compelling.

---

### Official Review · ~Justinas_Jučas3 · 2024-11-12
**Original Idea, but Requirements not Fulfilled**

**Rating:** 8
**Confidence:** 4

**Review:**

The proposal (which actually seems to be an already-implemented solution) provides a unique algorithm for a well-established and relevant text-to-SQL problem. The proposal contains all of the suggested sections in terms of the structure (although not in the identical way), contains a very clear algorithm. However, I am not sure why it also has an already-implemented solution of which the results are presented. Some cons also exist (described down below).
## Advantages
1. Very clear and well-structured algorithm. However, it is already tested and evaluated, so I am not sure what is the purpose of the proposal then.
2. Structure slightly differs from the proposed one, but all criteria are satisfied
3. The idea is very clear and original. In addition, text-to-SQL is a well-established and relevant problem.
## Disadvantages
1. The proposal length exceeds 2 pages, which was a strict requirement
2. Many grammatical mistakes (for instance, already in the title)
3. In the performance table, it is not clear what metric is used for evaluation

---

### Official Review · ~liyingxin1 · 2024-11-12
**Text-to-SQL may not be a very useful field in the future**

**Rating:** 7
**Confidence:** 3

**Review:**

Text-to-SQL is something that very few developers need to do, and in the future, large models may gradually revolutionize the middle-layer work, so I think choosing this topic is not necessarily particularly useful.

Apart from the above view, for this article, it is recommended to include a detailed analysis of the experimental results in the results section, particularly the performance differences of different methods on various evaluation metrics, and the underlying reasons for these differences.